# LPMARL: Linear Programming-based Implicit Task Assignment for Hierarchical Multi-agent Reinforcement Learning

## Abstract

Training a multi-agent reinforcement learning (MARL) model with sparse reward is notoriously difficult as the terminal reward is induced by numerous interactions among agents. In this study, we propose linear programming (LP)-based hierarchical MARL (LPMARL) to learn effective cooperative strategies among agents. LPMARL is composed of two hierarchical decision-making schemes: (1) solving an agent-task assignment LP using the state-dependent cost parameters generated by a graph neural network (GNN) and (2) solving low-level cooperative games among agents assigned to the same task. We train the LP parameter-generating GNN and the low-level MARL policy in an end-to-end manner using the implicit function theorem. We empirically demonstrate that LPMARL learns an optimal agent-task allocation and the subsequent local cooperative policy for agents in sub-groups for solving various mixed cooperative-competitive games.

## 1 Introduction

Multi-agent reinforcement learning (MARL) has recently drawn much attention due to its practical and potential applications in controlling complicated and distributed multi-agent systems. Despite its potential, training an MARL model with sparse reward is notoriously difficult as the final sparse reward is induced by the complex long-term interactions among the agents (Liu et al., 2021). To overcome this challenge, one needs to develop an algorithm that can learn how the interactions among the agents over long-term episodes entail the outcome of the target tasks, a delayed and sparse episodic reward, and deduce this understanding into an effective sequential decision-making policy.

In this study, we propose a linear programming-based hierarchical MARL (LPMARL), a hierarchically structured decision-making scheme, to learn an effective coordination strategy among the agents. LPMARL conducts two hierarchical decision-making: (1) solving an agent-task assignment problem and (2) solving local cooperative games among agents that are assigned to the same task. For the first step, LPMARL formulates the agent-task assignment as an LP by using the state-dependent cost coefficients generated by a graph neural network (GNN). The solution of the formulated LP serves as an agent-to-task assignment, which decomposes the original team game into a set of smaller team games among the agents that are assigned to the same task. LPMARL then employs a general MARL strategy to solve each sub-task cooperatively in the second step. We train the LP-parameter-generating GNN layer and the low-level MARL policy network in an end-to-end manner using the implicit function theorem. We validate the effectiveness of LPMARL using various cooperative games with constrained resource allocation.

The technical contributions and novelties of the proposed method are as follows:

- **Interpretability (Section 6.2.1).** LPMARL can induce the designed behavior of the agents (i.e., behavior inductive biases) by using specific objective terms or constraints when formulating the LP. This structured framework helps one to interpret the decision-making procedure.

- **Transferability (Section 6.2.2).** LPMARL learns to construct and solve the task assignment optimization problems. When constructing a resource assignment LP problem, LP-

MARL uses GNN to produce the state-dependent cost coefficient that will be used in the objective function of the LP. Due to the size generalization/transferability of GNN, the trained LPMARL policy can be used to solve general target problems with varying number of agents, tasks and constraints.

- **Amortization (Section 6.1)**. To execute LPMARL, the agent must solve the LP so as to use the optimal solution as the high-level action. As this centralized execution may be impossible in the real world, we amortize the central solution-finding procedure using a learned distributed task selection network. In the experiment, we demonstrate that the amortized high-level policy does not degrade the performance of the centralized version of LPMARL and outperforms existing hierarchical MARL algorithms.

## 2 BACKGROUND

### 2.1 HIERARCHICAL MULTI-AGENT REINFORCEMENT LEARNING

Tang et al. (2018) introduced a hierarchical Dec-POMDP that is composed of high- and low-level actions, which run on different time scales. This study introduced a temporal abstraction concept to integrate the two types of actions into the overall environmental dynamics. To be specific, each agent $i$ receives observation $o_{i,t}$ and chooses a high-level action $a_{i,t}^h \in \mathcal{A}_i^h$, where $\mathcal{A}_i^h$ denotes a set of possible high-level actions. While the high-level actions may last for $\tau$ timesteps, the low-level actions are executed until the current $a_{i,t}^h$ is terminated. After $a_{i,t}^h$ ends, the next high-level action $a_{i,t+\tau}^h$ is selected based on observation $o_{i,t+\tau}$. The agent receives a low-level reward (intrinsic reward) for reaching a sub-goal, denoted by $r^l(\boldsymbol{s}_t, \boldsymbol{a}_t, \boldsymbol{s}_{t+1} | a_{i,t}^h)$, depending to its own high-level action $a_{i,t}^h$. The agent receives the high-level reward $r^h(\boldsymbol{s}_t, \boldsymbol{a}_t, \boldsymbol{s}_{t+1})$ whenever the agent accomplishes the whole task, i.e., reaching the final success state $\boldsymbol{s}_{T+1}$.

### 2.2 IMPLICIT DEEP LEARNING

Implicit deep learning is a framework that incorporates implicit rules (e.g., ordinary differential equation (Chen et al., 2018), fixed-point iterations (Bai et al., 2019), and optimization (Amos & Kolter, 2017)) into a feed-forward neural network. Specifically, differentiable optimization is a framework that incorporates an optimization problem into the layer. A differentiable optimization layer takes the problem specific pamameters $x$ as an input, and finds optimal solution $z^*(x)$ such that $z^*(x) := \arg\min_{z \in g(x)} f(z, x)$, where $f(z, x)$ is the objective function constructed with a given $x$. The output of the layer, $z^*(x)$, is then fed into the next layer to conduct various end tasks.

By using this approach, one can infuse the optimization inductive bias into the layers. OptNet (Amos & Kolter, 2017), for example, propose a differentiable optimization layer, specifically for quadratic programming (QP). The backpropagation of this optimization layer requires the computation of the derivative of the QP solution with respect to the input parameters, which is derived by taking the matrix differentials of the KKT conditions of the QP. Ferber et al. (2020) and Wilder et al. (2019) extended the idea of OptNet to general LP and mixed-integer LP (MILP). To compute the gradient of the optimal solution in combinatorial optimization easily, Vlastelica et al. (2019) suggests a way to construct a continuous interpolation of the loss function.

## 3 RELATED WORK

**Hierarchical MARL with pre-defined high-level action space.** Some studies introduced hierarchical policies with pre-defined goal-level actions in MARL to decompose the main problem into subproblems under a semi-MDP framework (Sutton et al., 1999). Tang et al. (2018) applied a temporally abstracted high-level policy to induce the agents to cooperate when selecting the sub-goals. Ahilan & Dayan (2019) also introduced a centralized sub-goal selection policy to assign the agents to tasks optimally. Liu et al. (2021) introduced an exploration policy, which acts as a high-level policy, to limit the explorable action space of the low-level policy. Although these methodologies learn to divide the agents into goal-dependent sub-groups cooperatively, the low-level policies of these algorithms are trained individually, making it difficult to induce cooperation among the agents within the sub-tasks.

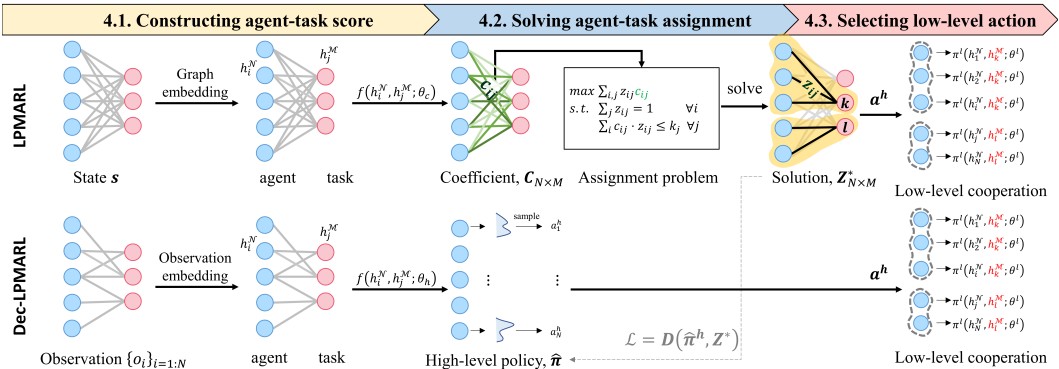

Figure 1: Overall decision-making framework of LPMARL

**Hierarchical MARL without pre-defined high-level action space.** Some studies employed a bi-level learning structure in MARL to learn the latent goals without pre-defining a high-level action set. Wang et al. (2020a) proposes to learn the latent goal of the agents by using the mutual information between the goals and the trajectories. Wang et al. (2020b) proposes to learn the latent representation of the action to cluster the joint action into the role action space. Yang et al. (2019) defines the low-level reward as a combination of team reward and intrinsic reward to balance the learning of the cooperation and individual skills. These methods indirectly limit the state space of the agents because the agents learn the policy that is conditioned on the latent embedding. Training the latent goal typically requires a lot of training loops with a dense-reward signal.

**Optimization-based agent-task decomposition.** Our work draws inspiration from Carion et al. (2019), a centralized optimization-based decomposition approach. In this paper, the high-level agent-task allocation problem is formulated into LP, where the optimization parameters are dynamically constructed depending on the state. By learning the state-dependent objective coefficient and constraint coefficient, the agents can be allocated optimally into tasks from a centralized point of view. However, as they use a rule-based low-level policy to train the high-level allocation policy, the agents may not behave cooperatively in the decomposed subproblem. Also, the optimization problems and low-level policy training are not end-to-end differentiable when trying to train two policies together. Our study is different from this in that we train the high- and low-level policies in an end-to-end manner while backpropagating the MARL loss through the LP optimization layer by using the implicit function theorem. In addition, we amortize the central LP-solving procedure with a decentralized task allocating policy, which allows each agent to choose the sub-goal cooperatively in a distributed manner, without having to solve the LP in the execution phase.

## 4 METHDOLOGY

At every step, LPMARL conducts three forward steps: (1) constructing an agent-task score matrix using GNN, (2) solving an agent-task assignment LP problem, and (3) solving local cooperative games among agents that are assigned to the same sub-task in parallel. Figure 1 shows the overall decision-making framework of LPMARL.

### 4.1 CONSTRUCTING AGENT-TASK SCORE MATRIX

We generate a state-dependent cost coefficient matrix $C \in \mathbb{R}^{N \times M}$ for the LP problem. First, given the global state $s = \{s_k : k \in \mathcal{N} \cup \mathcal{M}\}$ with $s_k$ being the features of node $k$, we construct a directed graph $\mathcal{G} = (\mathcal{V}, \mathcal{E})$ to describe the relationship between the agents and tasks explicitly. The graph nodes $\mathcal{V}$ are composed of agent nodes $\mathcal{N}$ and task nodes $\mathcal{M}$, i.e., $\mathcal{V} = \mathcal{N} \cup \mathcal{M}$. The agent node embedding and task node embedding are initialized as the local state of the agent and task, respectively. Two sets of edges are defined; agent-agent edges, $\mathcal{E}_{\mathcal{N}\mathcal{N}} = \{(i, j) : (i, j) \in \mathcal{N} \times \mathcal{N}\}$, and agent-task edges, $\mathcal{E}_{\mathcal{N}\mathcal{M}} = \{(i, j) : (i, j) \in \mathcal{N} \times \mathcal{M}\}$.

We use a message-passing GNN (Battaglia et al., 2018) to encode $\mathcal{G} = (\mathcal{V}, \mathcal{E})$:

$$m_{ij}^{\mathcal{NN}} \leftarrow f(s_i, s_j; \theta_g^{\mathcal{NN}}), (i,j) \in \mathcal{E}_{\mathcal{NN}}; \quad m_{ij}^{\mathcal{NM}} \leftarrow f(s_i, s_j; \theta_g^{\mathcal{NM}}), (i,j) \in \mathcal{E}_{\mathcal{NM}} \tag{1}$$

$$h_i \leftarrow f([\sum_{(i,j) \in \mathcal{E}_{\mathcal{NM}}} m_{ij} \| \sum_{(i,j) \in \mathcal{E}_{\mathcal{NN}}} m_{ij} \| s_i]; \theta_g^{\mathcal{V}}), \forall i \in \mathcal{V}s \tag{2}$$

where $\theta_g^{\mathcal{NN}}$ and $\theta_g^{\mathcal{NM}}$ are the parameters for the agent-agent and agent-task edge update functions, respectively; and $\theta_g^{\mathcal{V}}$ is parameter of the node update function. We denote the GNN-related parameters as $\theta_g = (\theta_g^{\mathcal{NN}}, \theta_g^{\mathcal{NM}}, \theta_g^{\mathcal{V}})$. After the graph encoding process, the updated node embeddings $\{h_i^{\mathcal{N}}\}_{i \in \mathcal{N}}$ for agents and $\{h_j^{\mathcal{M}}\}_{j \in \mathcal{M}}$ for tasks constructs an encoded graph, $\mathcal{G}' = (\mathcal{V}', \mathcal{E}')$.

Using the encoded graph node embeddings $\mathcal{V}'$, we can compute a coefficient matrix as follows.

$$c_{ij} := [C_{\theta_c}(\mathcal{G}')]_{i,j} = f(h_i^{\mathcal{N}}, h_j^{\mathcal{M}}; \theta_c) \tag{3}$$

where $\theta_c$ are the parameters of the network computing the agent-task scores, and the coefficient $c_{ij}$ is the allocation score for agent $i$ to finish task $j$. The constructed cost coefficients will then be used to construct the objective function of the agent-task allocation LP problem as shown in Eq. 4-7.

### 4.2 HIGH-LEVEL POLICY: SOLVING AGENT-TASK ALLOCATION OPTIMIZATION

**Centralized high-level policy.** We formulate the high-level agent-task assignment problem into an LP, especially for the resource-constrained allocation problem, as follows:

$$\text{maximize} \quad \sum_{i,j} c_{ij} \cdot z_{ij} \tag{4}$$

$$\text{s.t.} \quad \sum_i z_{ij} \leq k_j \qquad \forall j \in \mathcal{M} \tag{5}$$

$$\sum_j z_{ij} = 1 \qquad \forall i \in \mathcal{N} \tag{6}$$

$$0 \leq z_{ij} \leq 1 \qquad \forall i \in \mathcal{N}, j \in \mathcal{M} \tag{7}$$

where $\{z_{ij}\}_{i \in \mathcal{N}, j \in \mathcal{M}}$ is the decision variable that is to be optimized in a central way. Eq. 4 is the objective function that we want to maximize, Eq. 5 is the capacity constraint that restricts the maximal number of agents that can be assigned to each task, Eq. 6-7 are the constraints that ensures that the assignment probability of each agent is properly defined.

The optimal solution of the LP, $Z^* = \{z_{ij}^*\}_{i \in \mathcal{N}, j \in \mathcal{M}}$, can be considered as the output (assignments) of the high-level policy, i.e., $\pi^h(\cdot; \theta_h) : s \rightarrow Z^*$ where $z_{ij}^*$ is the probability that agent $i$ is assigned to task $j$. Agent $i$ then selects an action $a_i^h$ using this probability, i.e., $\pi^h(a_i^h = j|s) = z_{ij}^*$. Once every agent is assigned to one of the tasks, the agents are decomposed into sub-groups, each of which are composed of agents assigned to the same task. Formally, we denote the optimal solution $Z^* := Z^*(C_{\theta_c}(\mathcal{G}'))$ as a function of the input coefficients $C_{\theta_c}(\mathcal{G}')_{N \times M}$ produced by graph $\mathcal{G}'$.

**Decentralized high-level policy.** LPMARL has a centralized task assignment scheme as LP should be solved during both the training and execution phase. As the centralized execution may not be feasible during the execution level, we propose **Dec-LPMLARL**, a decentralized version of LPMARL, to assign each agent to a task in a decentralized manner without solving LP centrally.

To this end, in Dec-LPMARL, each agent's decentralized policy amortizes the optimum task allocation of LPMARL as $\hat{\pi}_i^h(a_i^h = j|o_i) = \hat{z}_{ij}^*$, where $\hat{\pi}_i^h$ is the GNN-based prediction function that maps the local observation $o_i$ to the action selection probability $\hat{z}_{ij}^*$ of agent $i$. Note that this imitation policy still considers the relationships among tasks and agents captured by the local observation $o_i = o_i(s)$. We use the shared decentralized policy $\hat{\pi}_1^h = \hat{\pi}_i^h = \cdots = \hat{\pi}_N^h = \hat{\pi}^h$ and $\hat{\pi}^h$ is trained to minimize the cross entropy between $Z^*$ and $\hat{Z}^*$. Figure 1 shows how each agent makes its high-level assignment action and coordinates with other agents in a decentralized manner with Dec-LPMLARL.

### 4.3 Low-level policy: Selecting low-level action

Once the global problem is decomposed into sub-tasks, the agent-task assignment is maintained for multiple timesteps until one of subtasks is finished. While the high-level action is fixed, each agent executes the low-level action in a decentralized manner every timestep to accomplish the assigned task cooperatively. For example, given the node embedding $h_i^{\mathcal{N}}$ of agent $i$ and the embedding $h_{a_i^h}^{\mathcal{M}}$ of the selected high level task $a_i^h$, agent $i$ selects the low level action $a_i^l$ as:

$$a_i^l \sim \pi^l(\cdot|o_i, a_i^h) = f(h_i^{\mathcal{N}}, h_{a_i^h}^{\mathcal{M}}; \theta_l). \tag{8}$$

where $\theta_l$ is the parameters of the low-level policy.

## 5 Training

We train the parameters $\theta_h$ of the high-level policy and the parameters $\theta_l$ of the lower-level policy at the same time by using the weighted sum of the high-level objective $\mathcal{J}(\theta_h)$ and the lower-level objective $\mathcal{J}(\theta_l)$:

$$\mathcal{J} = w \cdot \mathcal{J}(\theta_l) + (1 - w) \cdot \mathcal{J}(\theta_h) \tag{9}$$

We observed that high-level policies are harder to train than low-level policies at the beginning of the training because high-level rewards are more sparse than low-level rewards. Thus, to stabilize the training procedure, we schedule the varying of $w$. We set $w = 0.9$ initially and decayed linearly to $w = 0.1$. Ways for computing $\nabla_{\theta_h} \mathcal{J}(\theta_h)$ and $\nabla_{\theta_l} \mathcal{J}(\theta_l)$ is as following.

### 5.1 Actor and critic for the high-level policy

**Critic.** The high-level critic $Q^h(\cdot; \phi_h)$ is trained to minimize the the loss $\mathcal{L}(\phi_h)$, defined as:

$$\mathcal{L}(\phi_h) = \mathbb{E}_{(\boldsymbol{s}, a_i^h, r^h, \boldsymbol{s}') \sim D}\left[\left(r_t^h + \sum_i \gamma^\tau \cdot \max_{a_{i,t}^h} Q^h(\boldsymbol{s}_{t+\tau}, a_{i,t}^h; \bar{\phi}_h) - Q^h(\boldsymbol{s}_t, a_{i,t}^h; \phi_h)\right)^2\right] \tag{10}$$

where $\bar{\phi}_h$ is the target parameter. Since the same high-level action can be maintained for multiple timesteps, the target of the high-level critic is discounted proportionally to the duration when the high-level action remains the same.

**Policy Gradient.** The gradient of the high-level policy, $\nabla_{\theta_h} \mathcal{J}(\theta_h)$, can be expressed as:

$$\nabla_{\theta_h} \mathcal{J}(\theta_h) = \mathbb{E}_{(\boldsymbol{s}, a_i^h, r^h, \boldsymbol{s}') \sim D}\left[\sum_i Q^h(\boldsymbol{s}_t, a_{i,t}^h; \bar{\phi}_h) \cdot \nabla_{\theta_h} \log \pi^h(a_{i,t}^h|\boldsymbol{s}_t; \theta_h)\right] \tag{11}$$

$$= \mathbb{E}_{(\boldsymbol{s}, a_i^h, r^h, \boldsymbol{s}') \sim D}\left[\sum_i Q^h(\boldsymbol{s}_t, a_{i,t}^h; \bar{\phi}_h) \cdot \nabla_{\theta_h} \log z_{i,a_{i,t}^h}^*(\boldsymbol{s}_t; \theta_h)\right] \tag{12}$$

The combined procedures of (1) computing the node embedding, (2) constructing the agent-task score matrix $C$, and (3) solving the agent-task allocation problem can serve as the high-level policy. This decision-making pipeline constructs the following forward propagation chain:

$$\pi^h(\cdot; \theta_h) : \boldsymbol{s}_t \xrightarrow{\theta_h} C \xrightarrow{\text{solver}} Z^* \xrightarrow{\text{loss}} \mathcal{J} \tag{13}$$

Then, $\nabla_{\theta_h} \mathcal{J}(\theta_h)$ can be computed as the following chain rule:

$$\nabla_{\theta_h} \mathcal{J}(\theta_h) = \frac{\partial \mathcal{J}}{\partial Z^*} \cdot \frac{\partial Z^*(C)}{\partial C} \cdot \frac{\partial C}{\partial \theta_h} \tag{14}$$

In our optimization problem, there exists some extreme points generated only by the boundary constraints in Eq. 7. Thus, the optimal solution $Z^*(C)$ can be an integer value at these extreme points (please refer to Appendix A for a detailed description). In this case, the derivative $\frac{\partial Z^*(C)}{\partial C}$ can be in the form of the piecewise constant, making it difficult to estimate the gradient. Vlastelica et al. (2019) proposed to approximate the differentiation of integer programming layer to obtain a

meaningful gradient. The partial gradient $\frac{\partial \mathcal{J}}{\partial Z^*} \cdot \frac{\partial Z^*(C)}{\partial C} = \frac{\partial \mathcal{J}}{\partial C}$ is approximated by the solution gap between solution $Z^*$ of the original LP and solution $Z^*_\lambda$ of the perturbed LP:

$$\frac{\partial \mathcal{J}}{\partial C} \approx \frac{1}{\lambda}(Z^*_\lambda - Z^*) \tag{15}$$

The perturbed optimal solution $Z^*_\lambda$ is constructed by injecting the perturbed cost into the objective function of the LP (Eq. 4). This equation can be interpreted as an approximation of the tangent of a non-differentiable function using the slope between two discrete points. Please refer to Appendix B for more details on the backward computation of the optimization layer.

## 5.2 Actor and critic for the lower-level policy

**Critic.** The objective of the low-level policy is to maximize the sum of the individual low-level rewards assigned to a same task. Using the value function factorization (Sunehag et al. 2017) , the low-level critic $Q^l(\cdot; \theta_l)$ is trained to minimize the loss $\mathcal{L}(\phi_l)$, defined as:

$$\mathcal{L}(\phi_l) = \mathbb{E}_{\boldsymbol{s}, a_i^l, r^l, \boldsymbol{s}' \sim D}\left[\sum_{j \in \mathcal{M}}\left(r_j^l + \sum_{i, a_i^h = j} \gamma \cdot \max_{a_i^l} Q^l(\boldsymbol{s}', a_i^l; \bar{\phi}_l) - Q^l(\boldsymbol{s}, a_i^l; \phi_l)\right)^2\right] \tag{16}$$

where $r_j^l$ is team reward of sub-group $j$, and $\bar{\phi}_l$ is the target parameter.

**Policy gradient.** The low-level policy $\pi^l(\cdot|\boldsymbol{s}, a_i^h) = f(h_i^{\mathcal{N}}, h_{a_i^h}^{\mathcal{M}}; \theta_l)$ is then trained with the policy gradient objective

$$\nabla_{\theta_l}\mathcal{J}(\theta_l) = \mathbb{E}_{(\boldsymbol{s}, a_{i}^l, r_i^l, \boldsymbol{s}') \sim D}\left[\sum_i Q^l(\boldsymbol{s}_t, a_{i,t}^l; \bar{\phi}_l) \cdot \nabla_{\theta_l} \log \pi^l(a_{i,t}^l|\boldsymbol{s}_t; \theta_l)\right] \tag{17}$$

## 6 Experiment

In this section, we conduct several experiments to answer the following questions: (1) can LPMARL learn cooperative agent-task allocation, (2) is LPMARL transferable to different problem sizes, and (3) can Dec-LPMARL amortize the centralized task allocation optimization procedure using a decentralized imitation policy.

We compare LPMARL with the following baseline algorithms: (1) **Cooperative MARL algorithms**: Qmix (Rashid et al., 2018), MADDPG (Lowe et al., 2017), and MAAC (Iqbal & Sha, 2019), (2) **Hierarchical MARL**: HSD (Yang et al., 2019), RODE (Wang et al., 2020b) and CMAE[1](Liu et al., 2021).

In addition, we consider the two ablations of LPMARL: (1) **Dec-LPMARL** to check the feasibility of decentralizing LPMARL, (2) **No-LP** to observe the effectiveness of using LP as the high-level policy. The network architecture of No-LP and Dec-LPMARL are the same as LPMARL.

## 6.1 Cooperative navigation

The first environment is cooperative navigation, whose objective is to learn a one-to-one mapping of the agent-landmark (high-level action) as well as a navigation action (low-level action) to occupy all the landmarks (Lowe et al., 2017). We consider both dense- and sparse-reward settings. In the dense-reward setting, the reward is determined by the distance between the agent and the landmarks, the collision between agents, and the number of occupied landmarks. In the sparse-reward setting, an agent is rewarded only when the agent occupies the assigned landmark $(+1)$. In both reward settings, the agents receive a high-level reward when the agents occupy all the landmarks $(+1)$.

---

[1]Although CMAE does not use policies with a hierarchical level, it trains an exploration policy on the shrunk region of the full state space, which resembles the decomposition procedure by high-level policy of the hierarchical MARL.

The environment rewards are provided differently to the algorithms. For the hierarchical MARL algorithms and LPMARL, each level of reward described above is used to train each policy level accordingly. For the non-hierarchical MARL algorithms, a weighted sum of the high- and low-level rewards is used to train the policy. Additional experimental details are described on Appendix C.1.

Table 1: Success ratio (%) of cooperative navigation. The mean and standard deviation of 100 evaluation episodes are reported.

| Scenario | Ours | | | Non-hierarchical MARL | | | Hierarchical MARL | | |
|---|---|---|---|---|---|---|---|---|---|
| | LPMARL | Dec-LPMARL | No-LP | Qmix | MADDPG | MAAC | HSD | RODE | CMAE* |
| Dense-3 | **99.7**±0.5 | 89.1±1.1 | 78.3±1.5 | 82.4±1.5 | 85.7±1.1 | 89.1±3.1 | 98.9±0.9 | 99.1±0.8 | 98.2±1.4 |
| Dense-5 | **98.3**±1.5 | 84.5±1.2 | 44.5±5.3 | 62.0±1.4 | 21.3±5.2 | 74.5±4.6 | 97.3±1.6 | 98.2±1.3 | 84.2±3.6 |
| Dense-7 | **97.9**±1.4 | 85.7±2.1 | 56.2±4.9 | 49.1±5.2 | 5.7±0.1 | 65.7±4.7 | 96.9±1.7 | 95.8±2.0 | - |
| Sparse-3 | **90.0**±1.1 | 82.1±1.5 | 61.0±3.2 | 31.5±3.2 | 11.8±1.1 | 49.1±4.9 | 85.5±3.5 | 89.5±3.4 | 64.9±4.7 |
| Sparse-5 | **89.1**±2.9 | 74.9±2.0 | 49.2±4.5 | 32.4±4.2 | 8.0±1.3 | 39.5±4.8 | 80.9±3.9 | 76.1±3.2 | 49.0±4.8 |
| Sparse-7 | **86.5**±3.4 | 72.4±2.1 | 42.0±4.9 | 12.8±2.4 | 6.2±1.0 | 22.1±4.1 | 72.1±4.8 | 70.6±4.6 | - |

\* It is impossible to train CMAE on $N = 7$ due to the memory allocation problems, as CMAE considers a joint observation space of all agents ($|\mathcal{O}_i|^N$ dimensional) when training the exploration policy.

Table 1 shows the average performance metric of the different algorithms where *Dense-N* and *Sparse-N* refer to the dense and sparse settings with $N$ agents, respectively. Non-hierarchical MARL algorithms fail even in simple scenarios for both the dense and sparse reward settings. In the sparse reward setting, the success ratio of non-hierarchical algorithms is less than 50%. This shows that the general MARL algorithm cannot learn cooperation only with goal-oriented and sparse reward.

Hierarchical MARL algorithms succeed in every dense-rewarded setting. However, for the sparse-reward setting, the performance of hierarchical MARL methods degrades when $N$ increases. Unlike the other hierarchical MARL algorithms, LPMARL shows the smallest performance degradation when $N$ increases. If LPMARL successfully allocate task without conflict, it can solve the lower-level problem easily because the complexity of the decomposed sub-problem is the same as the complexity of the single-agent navigation problem.

No-LP has 20~40% lower performance than LPMARL on average, which indicates the effectiveness of the LP-based high-level task allocation. On the other hand, Dec-LPMARL has a slightly lower performance than LPMARL but has a considerably higher performance than No-LP, which verifies the potential of amortizing the central task allocation with a decentralized task allocation policy.

## 6.2 Constrained cooperative navigation

To further investigate the high-level assignment of LPMARL, we produced a modified version of the cooperative navigation environment, namely the constrained cooperative navigation. In this environment, $N$ agents aim to occupy $M$ ($M < N$) landmarks, where each landmark has its own capacity limit to accommodate the agents. In this environment, the agents receive a success reward only in the case when the agents occupy the landmarks while satisfying the capacity constraint. Additional experimental details are described on Appendix C.2

Table 2: Normalized high- and low-level reward of constrained cooperative navigation. The reported reward is the mean over all the agents over five runs.

| Scenario | | Ours | | | Non-hierarchical MARL | | | Hierarchical MARL | | |
|---|---|---|---|---|---|---|---|---|---|---|
| | | LPMARL | Dec-LPMARL | No-LP | Qmix | MADDPG | MAAC | HSD | RODE | CMAE |
| $(M, N) = (3, 5)$ | H | 0.86 | 0.73 | 0.33 | 0.37 | 0.21 | 0.27 | 0.49 | 0.43 | 0.35 |
| | L | 0.88 | 0.76 | 0.58 | 0.70 | 0.57 | 0.77 | 0.81 | 0.71 | 0.52 |
| $(M, N) = (5, 7)$ | H | 0.82 | 0.65 | 0.28 | 0.33 | 0.15 | 0.19 | 0.44 | 0.43 | 0.29 |
| | L | 0.89 | 0.66 | 0.47 | 0.73 | 0.52 | 0.78 | 0.82 | 0.62 | 0.59 |

Table 2 shows the normalized high- and low-level reward when the algorithm is trained with $(M, N) = \{(3, 5), (5, 7)\}$. The low-level reward can be interpreted as how close the agents are to the nearest landmark on average. Note that achieving maximum low-level reward does not imply that the algorithm has achieved success. For example, if the agents learn to navigate toward the landmark, they will get low-level rewards even if it is not an optimal landmark. The high-level reward can be interpreted as the success or failure of the algorithm. The figure shows that none of the baseline algorithms have learned to divide the agents into $M$ groups without violating the con-

straints. On the other hand, LPMARL learned an optimal allocation policy and a navigation policy, obtaining a 90% success ratio in the training environment. Additional experimental results including the training curves are reported on Appendix D.1.

### 6.2.1 INTERPRETATION OF HIGH-LEVEL POLICY

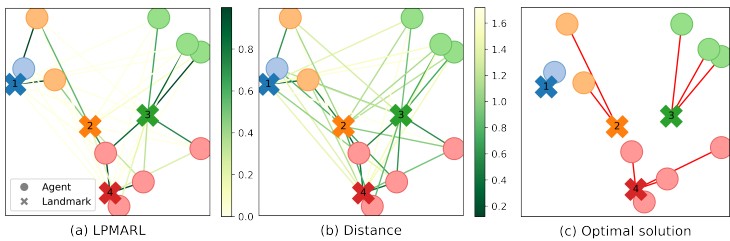

Figure 2: (a) Visualization of the learned coefficient of LPMARL. (b) Visualization of the agent-task distance matrix. (c) Optimal solution of LP with both input coefficients.

To interpret the learned optimization coefficient of LPMARL, we compare LPMARL with **LP-distance**, which uses the distance matrix when solving the upper-level assignment problem. Figure 2 shows the learned coefficient of LPMARL and distance for a single problem instance. By comparing Figure 2 (a)-(b), we can observe that the cost coefficient generating function $C_{\theta_c}(\cdot)$ generate cost coefficients that resembles the distance matrix. Although the learned coefficient function is sharper than the distance function, the optimal solution of LPMARL and theLP-distance is the same, as shown in Figure 2 (c). This experiment demonstrates how LPMARL performs task assignments by reflecting the relationship between agents and tasks. Although this task can be solved simply with distance matrices, LPMARL can work well while considering more complex relationships other than distance. Additional ablation study on high-level policy can be found in Appendix D.1.2

### 6.2.2 ZERO-SHOT POLICY TRANSFER

To see the size transferability of the LPMARL, we tested the algorithm in a constrained cooperative navigation environment with different sizes. However, since the input dimension of the baseline algorithm we consider in Section 6.2 has to be fixed, the trained baseline model cannot be transferred to a different problem with different number of agents. Therefore, we compared the performance only with our ablation model.

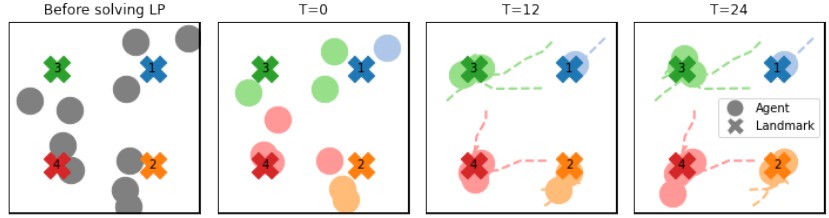

Figure 3: Visualization of transfer policy learned from $N = 5, M = 3$ to a task of $N = 10, M = 4$. The agent-landmark with the same color represents the result of the high-level assignment.

Table 3: Success ratio (%) of the transferring policy learned on $N, M \in [3, 10]$. The gray and blue shaded cell represents the in-training distribution and out-of-training distribution respectively.

| $N \rightarrow$ $M \downarrow$ | LPMARL | | | | | Dec-LPMARL | | | | | No-LP | | | | |
|---|---|---|---|---|---|---|---|---|---|---|---|---|---|---|---|
| | 3 | 5 | 10 | 15 | 20 | 3 | 5 | 10 | 15 | 20 | 3 | 5 | 10 | 15 | 20 |
| 3 | 98.4 | 96.5 | 85.2 | 72.2 | 66.2 | 93.5 | 88.1 | 68.1 | 40.7 | 32.1 | 51.2 | 30.9 | 9.2 | 0.0 | 0.0 |
| 5 | - | 98.2 | 82.1 | 70.3 | 62.8 | - | 79.5 | 54.8 | 41.0 | 29.2 | - | 35.2 | 5.8 | 0.0 | 0.0 |
| 10 | - | - | 79.5 | 65.2 | 49.1 | - | - | 54.0 | 22.5 | 11.5 | - | - | 3.2 | 0.0 | 0.0 |
| 15 | - | - | - | 68.9 | 53.2 | - | - | - | 17.5 | 8.9 | - | - | - | 0.0 | 0.0 |
| 20 | - | - | - | - | 48.0 | - | - | - | - | 5.7 | - | - | - | - | 0.0 |

The cost coefficient function and low-level policy, which are constructed based on GNN, can be used to solve games with different number of agents/tasks. To test its transferability, we transferred the policy learned from $N, M \in [3, 10](M \leq N)$ to a problem with a bigger size. Figure 3 shows an example behavior of LPMARL agent on $N = 10$, $M = 4$, $(k_1, k_2, k_3, k_4) = (1, 2, 3, 4)$. The text on the landmark indicates the capacity limit of each landmark. The leftmost subfigure in Figure 3 is the initial state of the environment, where the circles represent the agents and the cross marks represent the landmarks. In the second subfigure, the high-level policy assigns each agent to a landmark with the same color. After high-level assignment, the agents navigate toward the assigned landmark, resulting in the constraint-satisfied success state.

Table 3 shows the performance of the policy learned from $N, M \in [3, 10]$ that is tested on $N, M \in [3, 20]$ without further training. LPMARL still has over 50% success rate on out-of-training distribution. The LPMARL policy can transfer to a bigger scenario, showing its zero-shot transferability.

## 6.3 STARCRAFT2 MICROMANAGEMENT

The last environment we consider is the StarCraft Multi-Agent Challenge (SMAC, Samvelyan et al., 2019). In this environment, we can verify the dynamic goal assignment of the algorithms as the number of agents/enemies may change within the episode. The baselines we consider for SMAC are CMAE (Liu et al., 2021) and SEAC (Christianos et al., 2020), the MARL algorithms that are designed to solve cooperative tasks with sparse rewards. Note that other than CMAE and SEAC, there are no MARL algorithms that consider SMAC with sparse reward.

Table 4: Success ratio (%) on Starcraft Micromanagement environment. The mean and standard deviation over 100 evaluation episodes are reported.

| Map name | LPMARL | Dec-LPMARL | No-LP | Qmix | CMAE | SEAC | RODE |
|---|---|---|---|---|---|---|---|
| *3m_3m* (dense) | 98.4±1.4 | 97.1±1.2 | 85.1±5.3 | **100.0**±0.0 | 98.7±1.3 | 88.9±2.7 | **99.1**±0.7 |
| *2m_1z* (dense) | 99.7±0.5 | **99.1**±0.8 | 92.4±0.9 | **99.8**±0.1 | 98.2±0.9 | 87.5±2.8 | **99.8**±0.4 |
| *3s_5z* (dense) | 61.1±3.9 | 45.6±4.2 | 30.2±2.1 | 63.9±3.6 | 51.3±3.7 | 40.8±4.5 | **65.9**±3.4 |
| *8m_8m* (dense) | 92.2±1.4 | 80.4±2.3 | 72.5±3.1 | 95.2±1.7 | 84.2±1.9 | - | 90.2±1.2 |
| *3m_3m* (sparse) | **44.2**±3.7 | 31.6±2.0 | 8.4±1.3 | 10.4±3.0 | 42.2±4.8 | 8.3±2.8 | 18.6±2.1 |
| *2m_1z* (sparse) | **44.3**±4.9 | 21.4±3.0 | 5.4±0.6 | 13.4±3.3 | 17.2±3.8 | 30.5±4.6 | 35.1±4.7 |
| *3s_5z* (sparse) | **0.9**±0.5 | 0.0±0.0 | 0.0±0.0 | 0.0±0.0 | 0.1±0.0 | 0.0±0.0 | 0.3±0.0 |
| *8m_8m* (sparse) | 9.5±0.5 | 6.1±0.5 | 0.0±0.0 | 3.1±0.3 | **10.3**±0.9 | - | 2.6±0.2 |

Table 4 shows the final success ratio on SMAC tasks. In the dense-reward setting, LPMARL achieved at least similar or better performance than other baselines. Note that in *2m_1z*, LPMARL does not conduct the high-level action since there is only a single task; nevertheless, LPMARL shows good performances, possibly due to the effective state representation module constructed based on GNN.

In the sparse-reward setting, SEAC and RODE cannot learn a cooperative policy using only sparse rewards. On the other hand, SEAC and LPMARL achieve meaningful success ratios in *3m* and *2m_1z*. Note that LPMARL has a lower variance of success ratio than SEAC. In addition, although it is very low, LPMARL occasionally achieves a win for the difficult task *3s_5z*. Although it has a slightly lower success rate than its centralized version, Dec-LPMARL, it still has competitive performance compared to other baselines. On the other hand, No-LP fails to learn cooperative policy without using LP as a high-level policy. Additional results are reported on Appendix D.2

## 7 CONCLUSION AND LIMITATION

We proposed LPMARL to effectively solve cooperative/competitive game with sparse reward while optimally allocating agents to tasks. We demonstrated that LPMARL can decompose the agents into sub-tasks in various environments with the sparse-reward. Although LPMARL learns an effective allocation policy using the optimization layer, the training speed may be slower than conventional MARL algorithms with a feed-forward network because the batch computation of the optimization layer is impossible. Although our method has limitations in batched-training of the optimization layer, we demonstrated that the central optimization of task allocation can be amortized so that task allocation can be done in a decentralized manner.

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

## A  VISUALIZATION OF EXTREME POINT OF LP

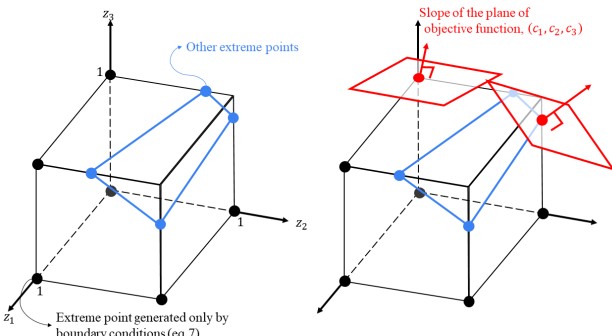

Figure 4: Visual description of extreme points

Figure 4 illustrates the extreme points of the linear programming in 3-dimensional case, where the decision variables are $(z_1, z_2, z_3) \in \mathbb{R}^3$. The boundary constraints are illustrated as the surface of the cube, and additional constraints other than boundary constraints are illustrated as the surface generated by blue lines. In Figure 4 (left), extreme points corresponding only to the boundary constraints are marked as the black dots, and the other extreme points are marked as the blue dots.

The objective of the optimization is to maximize $f(\boldsymbol{z}) = c_1 z_1 + c_2 z_2 + c_3 z_3$. The objective coefficients $(c_1, c_2, c_3)$ determine the slope of the plane of the objective function, as illustrated by the red arrow on Figure 4. In Figure 4, we consider two possible optimization outcomes. For case (1), the optimal solution $z^* = (z_1^*, z_2^*, z_3^*)$ occurs only on the extreme points that are generated by the boundary points, and for case (2): $z^*$ occurs on the other extreme points. In case (1), the optimal solution $z^*$ is integer-valued. In our LP formulation, we have $N \times M$ boundary constraints and $N + M$ additional constraints. Thus, depending on the value of the objective coefficient and constraint coefficients, there may exist some integer-valued solution. In addition, the solution of the LP hanges discontinuously on the continuous change on the slope of the objective plane. For this reason, we need continuous interpolation of the gradient of the discontinuous surface $\frac{\partial \mathcal{J}}{\partial C}$.

## B  POLICY GRADIENT APPROXIMATION FOR LP LAYER

The gradient of the high-level policy $\nabla_{\theta_h} \mathcal{J}(\theta_h)$ can be computed as the following chain rule:

$$\frac{\partial \mathcal{J}(\theta_h)}{\partial \theta_h} = \frac{\partial \mathcal{J}}{\partial Z^*} \cdot \frac{\partial Z^*(C)}{\partial C} \cdot \frac{\partial C}{\partial \theta_h} \tag{18}$$

We can compute the gradient $\frac{\partial Z^*(C)}{\partial \theta_h} = \frac{\partial Z^*(C)}{\partial C} \cdot \frac{\partial C}{\partial \theta_h}$ by differentiating the equality of the KKT conditions. Previous works (Wilder et al., 2019; Ferber et al., 2020) proposed to add additional quadratic regularization term $\gamma ||z||$ in the objective function (Eq. 4) to make LP as smooth QP, and induce non-singular Jacobian matrix to differentiate through the equality of the KKT conditions as in Amos & Kolter (2017).

However, although our high-level optimization problem is formulated as an LP, there exist extreme points generated only by the boundary constraints (Eq. 7). Thus, the optimal solution may be integer-valued on this extreme point as described in Appendix A. In this case, the optimal solution may not vary continuously with respect to the input; a small change can induce an abrupt change in the objective value. Thus, the $\frac{\partial Z^*(C)}{\partial C}$ be in the form of the piecewise constant, making it difficult to estimate the gradient. To solve this issue, Vlastelica et al. (2019) presented a piecewise linear surrogate loss surface that can approximate the original piecewise constant surface $\frac{\partial \mathcal{J}}{\partial C}$ of the combinatorial optimization layer. To obtain a meaningful differential value for this integer-solution, the gradient $\frac{\partial \mathcal{J}}{\partial C} = \frac{\partial \mathcal{J}}{\partial Z^*} \cdot \frac{\partial Z^*(C)}{\partial C}$ approximated by using the solution gap between the original solution of the optimization problem and the solution of the perturbed optimization problem, as:

$$\frac{\partial \mathcal{J}}{\partial C} \approx -\frac{1}{\lambda}(Z^* - Z_\lambda^*) \tag{19}$$

where $Z^*$ and $C$ are the optimal solution and coefficient used in the forward pass, respectively. $Z^*_\lambda$ is the approximate solution computed as:

$$Z^*_\lambda = Z^*(C'_\lambda). \tag{20}$$

Here, $C'_\lambda$ is the perturbed cost coefficient computed as

$$C'_\lambda := C + \lambda \cdot \frac{d\mathcal{J}}{dZ}(Z^*) \tag{21}$$

where $\lambda$ is the hyperparemeter that scales the amount of perturbation while considering the gradient with respect to the solution $Z^*$. $\frac{d\mathcal{J}}{dZ}(Z^*)$ is the gradient of $\mathcal{J}$ with respect to solver output $Z$ at given point $Z^*$. The amount of solution gap between the original solution and perturbed solution $Z^* - Z^*_\lambda$ is the slope of the piecewise linear function that will replace $\frac{\partial \mathcal{J}}{\partial C}$ evaluated at $C$. One can guarantee that the modified loss function is piecewise affine and similar to the original loss function (we refer to Vlastelica et al., 2019 for more detail). In the backward pass, Eq. 19 replaces $\frac{\partial \mathcal{J}}{\partial Z^*} \cdot \frac{\partial Z^*(C)}{\partial C}$ in Eq. 14.

## C  HYPERAPAREMETERS AND EXPERIMENTAL DETAILS

The hyperparameters of LPMARL used in the experiment are summarized in Table 5. We used an CVXPY solver (Diamond & Boyd, 2016) to solve the linear programming in Section 4. s

Table 5: Hyperparemeters of LPMARL

| Hyperparemter | Values |
|---|---|
| MLP units for GNN, $f(\cdot; \theta_g^{\mathcal{NM}}), f(\cdot; \theta_g^{\mathcal{MM}}), f(\cdot; \theta_g^V)$ | [32,32] |
| MLP units for coefficient matrix, $f(\cdot; \theta_c)$ | [64,64] |
| MLP units for policy network, $f(\cdot; \theta_h), f(\cdot; \theta_l)$ | [64,64] |
| MLP units for critic network $f(\cdot; \phi_h), f(\cdot; \phi_l)$ | [64,64] |
| Nonlinear activation | LeakyReLU, negative slope=0.01 |
| learning rate | $10^{-3}$ |
| Discount rate, $\gamma$ | 0.99 |
| $\lambda$ | 40 |
| Optimizer | Adam |

For other baselines algorithms, we used a two-layer feed-forward fully-connected network with a 64-dimensional hidden layer and ReLU activation. Batches of 32 episodes are sampled from the replay buffer. The optimizer, learning rate, and discounting rate $\gamma$ of the other algorithms are set to be the same as in Table 5. Experiments are carried out on NVIDIA RTX A6000 GPU.

### C.1  COOPERATIVE NAVIGATION

We used 50,000 episodes where the maximum timestep of the each episode is 50. For every algorithm, we set the size of the replay buffer as 5,000 with a batch size of 100 to train. For LPMARL, we set $k_i$, the capacity constraint coefficient of Eq. 5, as 1 at every scenario. The initial location of the agents and landmarks are randomly spawn on $[-1, 1] \times [-1, 1]$ and $[-0.8, 0.8] \times [-0.8, 0.8]$, respectively. The reward conditions of the experiment is specified in Table 6.

### C.2  CONSTRAINED COOPERATIVE NAVIGATION

We used 50,000 episodes where the maximum timestep of the each episode is 50. For LPMARL, we set $k_i$, the capacity constraint coefficient of Eq. 5, to be the same as the landmark capacity. The initial location of the agents and landmarks are randomly spawn on $[-1, 1] \times [-1, 1]$ and $[-0.8, 0.8] \times [-0.8, 0.8]$, respectively.

| Reward setting | | Hierarchical MARL, LPMARL | Non-hierarchical MARL |
|---|---|---|---|
| Dense setting | High-level reward | +1 when all agents reaches landmark | 0.5×High-level reward +0.5×Low-level reward |
| | Low-level reward | +1 when each agent reaches landmark distance and collision-related reward | |
| Sparse setting | High-level reward | +1 when all agents reaches landmark | 0.5×High-level reward +0.5×Low-level reward |
| | Low-level reward | +1 when each agent reaches landmark | |

Table 6: Reward setup of cooperative navigation environment.

We set the capacity of the landmarks as an integer partition of $N$ into $M$ groups, i.e., $\sum_{j=1}^{M} k_j = N$ where $k_j$ is the number of agents that can be accommodated by landmark $j$. The capacity of the landmark is randomly determined at the beginning of each episode. Agents can observe the capacity and the position of each landmark within the observation range.

### C.3 STARCRAFT MULTI-AGENT CHALLENGE

We train all models over 100,000 episodes. For every algorithm, we set the size of the replay buffer as 5,000 with a batch size of 100 to train. For LPMARL, we set $k_i$, the capacity constraint coefficient of Eq. 5, as $\lceil \frac{\mathcal{M}}{2} \rceil$ at every scenario.

We consider the following SMAC scenarios:

- *3m*: 3 marines ($N = 3$) versus 3 marines ($M = 3$). The episode limit is 60 timesteps.
- *2m_1z*: 2 marines ($N = 2$) versus 1 zealot ($M = 1$). The episode limit is 150 timesteps.
- *3s_5z*: 3 stalker ($N = 3$) versus 5 zealots ($M = 5$). The episode limit is 150 timesteps.
- *8m*: 8 marines ($N = 8$) versus 8 marines ($M = 8$). The episode limit is 150 timesteps.

Difficulty levels of the scenarios are all set to be *harder (6)*.

## D ADDITIONAL EXPERIMENT RESULTS

### D.1 COOPERATIVE NAVIGATION

#### D.1.1 TRAINING CURVE

Training curve on constrained cooperative navigation is shown in Figure 5. Figure 5 shows the high- and low-level reward curve over the episodes when the algorithm is trained when $N$ and $M = 3$.

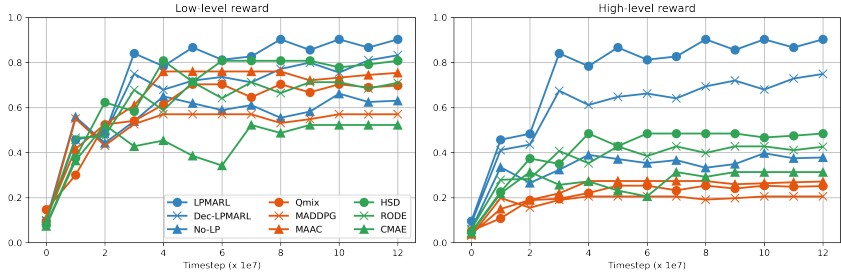

Figure 5: Low-level (left) and high-level (right) reward curve. The reported reward is the mean over all the agents over the timestep per episode.

#### D.1.2 EFFECTIVENESS OF LP LAYER

We consider the following ablations of high-level policy to examine the effectiveness of using LP as a high-level policy. **LP-distance** assigns an agent to a goal using the distance matrix as the objective

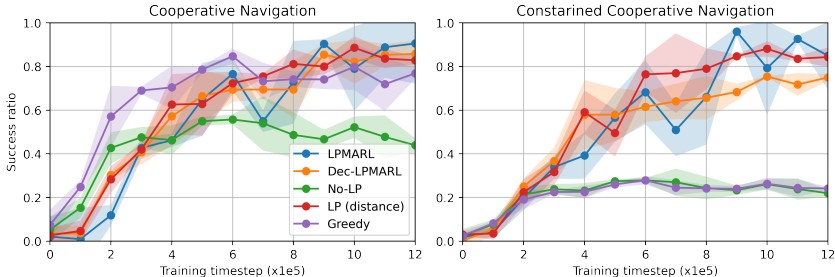

Figure 6: Training curve of ablations of high-level policy on cooperative navigation (left) and constrained cooperative navigation (right)

coefficient of LP. Therefore, agents are assigned to landmarks that minimize the sum of the distance of all the agents. **Greedy** is a greedy assignment, where each agent chooses the closest landmark individually. The network structure for low-level policies is identical for all the high-level ablations.

Figure 6 shows the results of ablation studies. LP-distance uses hand-designed features to induce good coordination for tasks where proximity plays essential roles, such as cooperative navigation. However, devising a hand-defined rule for a complex task is challenging. In such cases, the proposed algorithm that constructs the cost matrix considering the global state can play an important role in deriving an effective policy. By comparing LPMARL and No-LP, we can see that the LP layer plays a vital role in allocating agents to tasks to increase the success rate.

### D.1.3 EFFECT OF SPARSE REWARD HELP FOR TRAINING NON-HIERARCHICAL MARL

In cooperative navigation, although non-hierarchical MARL can be trained only with dense reward (distance and collision-related reward), we use a weighted sum of the high-level (sparse) and low-level reward to compare the performance fairly. If only dense reward is used for non-hierarchical MARL, their performance degrades. Table 7 compares the success ratio of using only dense reward for non-hierarchical MARL algorithms on cooperative navigation environment.

|  | Qmix | | MADDPG | | MAAC | |
|---|---|---|---|---|---|---|
|  | T+E | E | T+E | E | T+E | E |
| Dense-3 | 82.4 | 87.0 | 85.7 | 82.1 | 89.1 | 85.5 |
| Dense-5 | 62.0 | 60.9 | 21.3 | 32.8 | 74.5 | 76.2 |
| Dense-7 | 49.1 | 43.0 | 5.7 | 12.2 | 65.7 | 52.0 |

Table 7: Success ratio when cooperative MARL algorithms are only trained with environment reward. (T+E) indicates task-dependent reward + environmental reward, and (E) indicates environmental reward.

### D.2 SMAC ENVIRONMENT

Figure 7 visualizes the high-level assignment of LPMARL on *8m* environment. We divided the enemy units into two groups, with three and five units, to clearly see how LPMARL allocates agents to enemies. Each sub-figure of Figure 7 shows the high-level assignment result on timestep when the event (when $M$ or $N$ changes) occurs. The agent-task with the same color represents the result of the high-level assignment.

In the figure, we can observe that the LPMARL agent sequentially kills all the enemies by focusing fire from the nearest enemy (task). Also, LPMARL does not always assign only the closest enemy to the ally but assigns agents to the enemy with the lowest health level ($N = 5, M = 4$). Further visualization videos can be found at the following link.

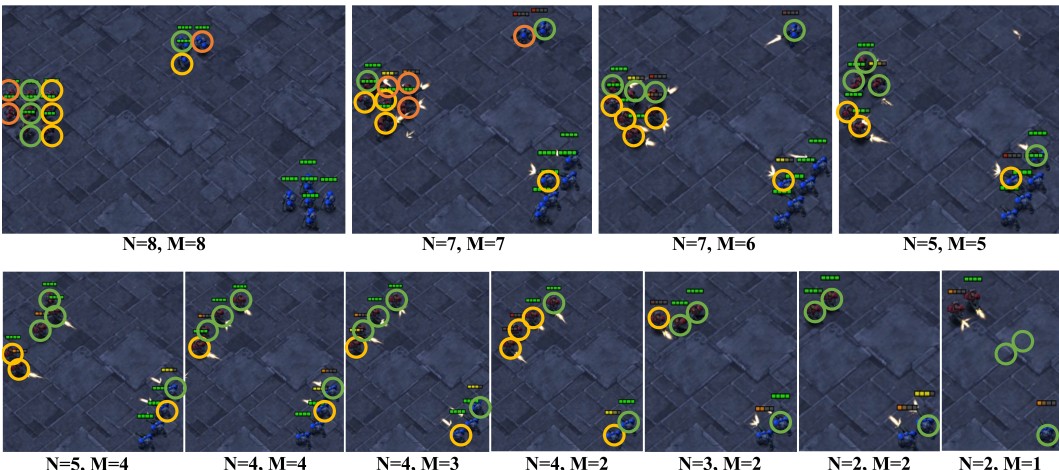

Figure 7: Visualization of high-level assignment on StarCraft environment (*8m*).

# E CODES

The code repository will be available after the decision.

