# OpenReview forum: "LPMARL: Linear Programming based Implicit Task Assignment for Hierarchical Multi-agent Reinforcement Learning"
_ICLR.cc/2023/Conference — Submitted to ICLR 2023_

### Official Review · Reviewer_LMn4 · 2022-10-23

**Confidence:** 3
**Correctness:** 3
**Technical Novelty And Significance:** 2
**Empirical Novelty And Significance:** 2
**Recommendation:** 5

**Clarity, Quality, Novelty And Reproducibility:**

The writing of the paper is OK and the idea of the paper is somehow understandable.

**Strength And Weaknesses:**

Strength

• The idea of leveraging techniques for differentiation of black box combinatorial solvers for MARL is unique and interesting.

• Experiments are extensive. Fig.1 and Fig.3 are the good visualizing explanation.

Weakness
main comments:
• what is the advantage of using a differentiable LP layer (GNN and a LP solver) as a high-level policy, shown in Eq. 10?

– compare it to [1] that considers the LP optimization layer as a meta-environment?

– compare it to an explicit task assignment protocol (e.g. not implicit).

E.g. a high-level policy that directly outputs task weightings instead of the intermediary C matrix?

• How does this method address sparse reward problems in a better way? From the experiments, this does not support well. in practice, the proposed method requires sub-task-specific rewards to be specified, which would be similar to providing a dense reward signal that includes rewards for reaching sub-goals. If given the sum of low-level reward as the global reward, will the other methods (Qmix) solve the sparse-reward tasks as well?

minor comments:
• It is hard to determine whether the solution to the matching problem (learned agent-task score matrix C) optimized by LP is achieving global perspective over the learning process.

• When the lower-level policies are also trained online, the learning could be unstable. Details on how to solve the instability in hierarchical learning are missing.

• What is the effect of the use of hand-defined tasks on performance? what is the effect of the algorithm itself? maybe do an ablation
study.

• Section 5.2 ”training low-level actor-critic” should be put in the main text.

[1] Carion N, Usunier N, Synnaeve G, et al. A structured prediction approach for generalization in cooperative multi-agent reinforcement learning[J]. Advances in neural information processing systems, 2019, 32: 8130-8140.


**Summary Of The Paper:**

This introduces a hierarchical framework for multi-agent reinforcement learning. It formulates agent-task assignments as a linear programming problem. With the solution of LP, it generates a low-level policy to solve each sub-task in a cooperative manner. Some empirical experiments are performed to demonstrate the effectiveness of the proposed method.

**Summary Of The Review:**

This paper is a resubmission of their work from last year.  Compared with last year's submission, the contribution and novelty are marginal.

---

> ### Author Response · Authors · 2022-11-15
> **Comment for the reviewer LMn4 (1)**
>
> > What is the advantage of using a differentiable LP layer (GNN and a LP solver) as a high-level policy, shown in Eq. 10?
>
> **Comparing to [1] using LP layer to solve agent-task allocation problem**
>
> Our method is similar to [1] in that both studies use LP to formulate state-dependent agent-task allocation problems and use the solution of the LP in allocating agents to tasks. The major difference between LPMARL and [1] is in the training procedure. [1] only trains the ‘scoring model’, which is used as the input parameter (objective coefficient and constraint coefficient of LP or QP), while treating low-level policy, environment, and optimization model (LP, QP, or argmax) as ‘meta-environment’.
>
> Our study differs from [1] in that we train the high- and low-level policies in an end-to-end manner while backpropagating the MARL loss through the LP optimization layer using the implicit function theorem. In this way, we could impose optimization inductive bias in the existing MARL framework. In addition, we amortize the central LP-solving procedure with a decentralized task-allocating policy, which allows each agent to choose the sub-goal cooperatively in a decentralized manner without having to solve the LP in the execution phase; however, [1] always needs to solve centralized optimization problem during execution.
>
> **Compare it to an explicit task assignment protocol (e.g. not implicit; a high-level policy that directly outputs task weightings instead of the intermediary C matrix?)**
>
> No-LP, the ablated version of LPMARL, uses explicit task allocation protocol without solving LP in a centralized manner for allocating agents to tasks. No-LP accepts the global state as an input and directly outputs the probability for agent-task allocation, which is used as high-level policy. For all three experiments, LPMARL outperforms the No-LP model, implying that agent-task allocation computed by solving LP in the layer is more effective in inducing cooperative task completion among agents than explicit task assignment.

---

> ### Author Response · Authors · 2022-11-15
> **Comment for the reviewer LMn4 (2)**
>
> > How does this method address sparse reward problems in a better way? From the experiments, this does not support well. in practice, the proposed method requires sub-task-specific rewards to be specified, which would be similar to providing a dense reward signal that includes rewards for reaching sub-goals. If given the sum of low-level reward as the global reward, will the other methods (Qmix) solve the sparse-reward tasks as well?
>
> A sub-task-specific reward is an indication type reward that is provided when an agent accomplishes a sub-task; thus, this reward is realized in a sparse manner. On the other hand, a dense reward is provided whenever the action is taken with the state transition; thus, the dense reward should be carefully designed to induce an agent to achieve the desired tasks.
>
> Despite the fundamental difference, we agree that specifying sub-task-specific rewards would be similar to providing a dense reward signal when agents accomplish a sub-goal. In that sense, non-hierarchical MARL algorithms such as MADDPG and Qmix can still be trained with such sub-task-specific rewards. We have already conducted experiments suggested by the reviewer. We train Non-hierarchical MARL algorithms (Qmix, MADDPG, MAAC) using the sub-task-specific reward to solve the sparse-reward tasks for all three experiments. The following tables show the reward setups for such experiments.
>
> (Table 6, Appendix C.1.)
> | Reward setting |            | Hierarchical MARL, LPMARL                                                   | Non-hierarchical MARL                       |
> |----------------|------------|-----------------------------------------------------------------------------|---------------------------------------------|
> | Dense          | high-level | +1 when all agent reaches landmark                                          | 0.5* high-level reward+0.5*low-level reward |
> |                | low-level  | +1 when each agent reaches landmark + distance and collision-related reward |                                             |
> | Sparse         | high-level | +1 when all agent reaches landmark                                          | 0.5* high-level reward+0.5*low-level reward |
> |                | low-level  | +1 when each agent reaches landmark                                         |                                             |
>
> As shown in Table 1 and Table 2, LPMARL and Dec-LPMARL always perform better than non-hierarchical MARL algorithms. From these results, defining sub-tasks and providing sub-task-specific rewards properly to non-hierarchical MARL is not good enough to induce effective cooperation among agents. We believe that LPMARL performs better in sparse reward settings because LPMARL considers the agent-task relationships and explicitly solves agent-task allocation optimization problems in determining the lower-level action. On the other hand, non-hierarchical MARL should learn such optimum allocation through training, which is not easy.
>
> To be more specific, we decompose the whole task into higher-level assignments and low-level local coordination among agents. We then have an LP layer to conduct the higher-level tasks and MARL to conduct the lower-level coordination tasks. We believe that this effective division of roles contributed to performance improvement. Note that although we divided the decision-making roles, we trained these two modules simultaneously in an end-to-end manner so that these two modules work together to increase the overall performance.

---

> ### Author Response · Authors · 2022-11-15
> **Comment for the reviewer LMn4 (3)**
>
> > It is hard to determine whether the solution to the matching problem (learned agent-task score matrix C) optimized by LP is achieving global perspective over the learning process.
>
> The cost-generating function $c_{ij}=C_{\theta_c}(g)$  accepts the graph g constructed from the global state s as an input and predicts the cost matrix element. The graph state captures the relative relationships among agents and tasks, and GNN layer effectively extracts the essential relationships among them to make the optimal agent-task allocation problems from a global perspective.
>
> ---
>
> > When the lower-level policies are also trained online, the learning could be unstable. Details on how to solve the instability in hierarchical learning are missing.
>
> Thank you for pointing out the missing details regarding the trading procedure. We train all the parameters by minimizing the global loss L, a weighted sum of the upper-level loss $J^h (\theta_h)$ and the lower-level loss $J^l (\theta_l)$
>
> $J=wJ^l (\theta _l )+(1-w)J^h (\theta_h)$
>
> We achieve the learning stability by adjusting the weight w over the training. At the beginning of the training, high-level policies are harder to train than low-level policies because high-level rewards are more sparse than low-level rewards. Thus, we set $w=0.9$ initially and decayed linearly to $w=0.1$ ($-2e-5$ at every training step). We included the description of the training details on the main text (Section 5).
>
> ---
>
> > What is the effect of the use of hand-defined tasks on performance? what is the effect of the algorithm itself? maybe do an ablation study.
>
> The high-level policy of LPMARL is composed mainly of two parts: (1) cost coefficient generating function producing the cost matrix to be used by LP agent-task allocation problem considering the relationships among agents and task, and (2) finding the solution of LP to optimally allocate agents to tasks.
>
> To evaluate the effectiveness of (1), we have compared LPMARL and LP (distance), the model using the cost coefficient matrix computed solely based on the distance between agents and tasks.
>
> To evaluate the effectiveness of (2), we have compared LPMARL and No-LP, the model that does not use LP in generating the agent-task allocation results but uses just the GNN layer to produce assignment probability.
>
> Figure 6 in Appendix D.1.2 shows the results of these ablation studies. LP (distance) using hand-designed features induces good coordination for tasks where proximity plays essential roles, such as cooperative navigation. However, devising a hand-defined rule for a complex task is challenging. In such cases, the proposed algorithm that constructs the cost matrix considering the global state can play an important role in deriving an effective policy. By comparing LPMARL and No-LP, we can see that the LP layer plays a vital role in allocating agents to tasks to increase the success rate.
>
> ---
> > Section 5.2 ”training low-level actor-critic” should be put in the main text.
>
> We have included the description to train the lower-level policy into Section 5.2 in the revised manuscript.

---

### Official Review · Reviewer_VRWi · 2022-10-23

**Confidence:** 4
**Correctness:** 3
**Technical Novelty And Significance:** 2
**Empirical Novelty And Significance:** 2
**Recommendation:** 5

**Clarity, Quality, Novelty And Reproducibility:**

**Clarity:** The paper needs improvement with using consistent notations throughout the paper.
**Quality & Novelty:** I agree that the high-level task allocation part is novel.
**Reproducibility:** The source code is not provided, so it is not trivial to reproduce the results.

**Strength And Weaknesses:**

**Strengths:**
1. Formulating the high-level agent-task allocation problem as LP is novel.
2. The interpretability and transferability are beneficial properties of LPMARL.

**Weaknesses and Concerns:**
1. My biggest concern is possible unfair comparisons against baselines. In particular, LPMARL is centralized training & centralized execution approach, and the baselines are centralized training & decentralized execution approach. Therefore, comparing Dec-LPMARL (centralized training & decentralized execution) against the baselines for fairness, it is unclear whether Dec-LPMARL is a clear winner (e.g., HSD performs better than Dec-LPMARL in Table 1).
2. The clarity of the paper generally needs improvement. For example, this paper uses (hierarchical) Dec-POMDP as the base framework, so each agent receives observation $o_{i,t}$ not the state $s_t$. However, LPMARL uses the state information (see Figure 1, where the input is the state) to construct the agent-task score, and it is unclear how the state can be constructed based on observations. Similarly, the paper uses $s_i$, but it should be $o_{i}$. Lastly, Equation 8 uses the state to decide the low-level action, but it should be $o_i$.
3. Regarding the non-hierarchical MARL baselines, it is unclear why the weighted sum of the high- and low-level rewards is used to train them. Wouldn't it be fairer to train them with the environment rewards only?

**Minor:**
1. In Figure 1, instead of "ag", "agent" can improve readability.
2. Typo: "Dec-LPMLARL" to "Dec-LPMARL"
3. Could you clarify what high-level action space is in the cooperative navigation domain?

**Summary Of The Paper:**

This paper introduces the LPMARL framework to address the sparse reward challenge in MARL. Specifically, LPMARL consists of two hierarchical decision-making parts: 1) the high-level part solves the agent-task assignment as the LP problem, and 2) the low-level part solves local games among agents with the same task. To enable decentralized execution, the paper also develops Dec-LPMARL by amortizing the task allocation of LPMARL. Empirical evaluations in cooperative navigation and SMAC domains show the effectiveness of the proposed approach.

**Summary Of The Review:**

I would like to initially vote for 5 due to the concerns above. After the authors' response to my concerns and questions, I will make a final decision on the recommendation.

---

> ### Author Response · Authors · 2022-11-15
> **Comment for the reviewer VRWi (1)**
>
> > My biggest concern is possible unfair comparisons against baselines. In particular, LPMARL is centralized training & centralized execution approach, and the baselines are centralized training & decentralized execution approach. Therefore, comparing Dec-LPMARL (centralized training & decentralized execution) against the baselines for fairness, it is unclear whether Dec-LPMARL is a clear winner (e.g., HSD performs better than Dec-LPMARL in Table 1).
>
> We initially proposed LPMARL to simultaneously conduct task assignment and lower-level control, a known notoriously difficult task. However, after knowing that the MARL community has devoted enormous effort to solve this task in a decentralized execution manner, we determined to develop a decentralized version of LPMARL to mimic the powerful decentralized execution capability of MARL algorithms. Thus, we agree with the reviewer’s opinion that Dec-LPMARL should be compared with other baselines designed for decentralized execution.
>
> In the first experiment (Cooperative Navigation), HSD performs slightly better than Dec-LPMARL. However, in the second environment (Constrained Cooperative Navigation), Dec-LPMARL performs better than HSD. The high-level reward (success rate of task) of Dec-LPMARL and HSD are, respectively, 0.73 and 0.49 for (M=3, N=5), which means that Dec-LPMARL has a 24% higher success probability than HSD. In addition, the high-level reward (success rate of task) of Dec-LPMARL and HSD are, respectively, 0.65 and 0.44 for (M=5, N=7), which means that Dec-LPMARL has a 19% higher success probability than HSD. To clearly show these quantitative results, we have inserted the following table in the revised manuscript (Section 6.2.). The original training curves are now moved to Appendix D. 1.1
>
> (Table 2 on section 6.2, normalized high-and low-level reward of constrained cooperative navigation)
> |  scenario |                   |  ours  |            |       | Non-hierarchical MARL |        |      | Hierarchical MARL |      |      |
> |:---------:|:-----------------:|:------:|------------|-------|-----------------------|--------|------|-------------------|------|------|
> |           |                   | LPMARL | Dec-LPMARL | No-LP | Qmix                  | MADDPG | MAAC | HSD               | RODE | CMAE |
> | M, N=3, 5 | High-level reward |   .86  | .73        | .33   | .37                   | .21    | .27  | .49               | .43  | .35  |
> |           |  low-level reward | .88    | .76        | .58   | .70                   | .57    | .77  | .81               | .81  | .52  |
> | M, N=5, 7 | High-level reward | .82    | .65        | .28   | .33                   | .15    | .19  | .44               | .43  | .29  |
> |           |  low-level reward | .89    | .66        | .47   | .73                   | .52    | .78  | .82               | .62  | .59  |
>
> We believe that once a more complex constraint is imposed in task allocation, the general hierarchical MARL algorithm suffers difficulty in learning optimally allocating limited resources to tasks only based on purely model-free MARL with representation learning. In such cases, introducing behavior inductive bias by solving a task allocation optimization problem in a layer (or amortizing the optimal allocation solution) is beneficial.

---

> > ### Comment · Reviewer_VRWi · 2022-11-17
> > **Response to Rebuttal**
> >
> > I appreciate the authors for the clarifications, additional experiments, and changes to the paper. The response and changes have addressed my concerns for #2 (i.e., state clarification) and #3 (i.e., non-hierarchical MARL baselines). However, my concern #1 (i.e., Dec-LPMARL performance) remains because Dec-LPMARL's performance is similar to or lower than its competitive decentralized hierarchical baselines (CMAE, RODE; HSD baseline is not shown). Hence, I would like to maintain my score.

---

> > > ### Author Response · Authors · 2022-11-22
> > > **Comment for the reviewer VRWi**
> > >
> > > > I appreciate the authors for the clarifications, additional experiments, and changes to the paper. The response and changes have addressed my concerns for #2 (i.e., state clarification) and #3 (i.e., non-hierarchical MARL baselines). However, my concern #1 (i.e., Dec-LPMARL performance) remains because Dec-LPMARL's performance is similar to or lower than its competitive decentralized hierarchical baselines (CMAE, RODE; HSD baseline is not shown). Hence, I would like to maintain my score.
> > >
> > > First of all, we sincerely appreciate the reviewer for providing valuable feedback on our work.
> > >
> > > ---
> > > **Why authors did not include HSD in the SMAC environment?**
> > >
> > > We couldn't train the HSD in the SMAC environment because the algorithm contains a lot of environment-related hyperparameters, such as high-level skill maintaining timesteps, the proportion of segments used for skill discovery, the number of skills, and so on. The hyperparameter settings were obvious for the cooperative navigation environment; however, it did not work on sc2 using only the open-source code of HSD.
> > >
> > > ---
> > > **The performance of Dec-LPMARL**
> > >
> > > We agree that Dec-LPMARL does not outperform existing hierarchical MARL algorithms in the SMAC environment. However, we verified that Dec-LPMARL indeed overperforms the hierarchical MARL algorithm in a constrained cooperative navigation environment, a complex cooperative task with hard constraints imposed on. For example, even in a SMAC environment, one may have prior knowledge of combat, such as "at least two marines are required to defeat a single Zealot" or "allocating more than five marines are waste to defeat a single Stalker." LPMARL and Dec-LPMARL can impose such optimization constraints into the framework, whereas existing works cannot. Please note that we did not impose such constraints on SMAC currently. We believe that if those constraints are added to the intermediate LP layer, Dec-LPMARL can further improve.
> > >
> > > Painfully, we have to agree with the reviewer's concern about the performance of Dec-LPMARL is not yet impressive enough. Thanks to the reviewer's opinion, however, we have come up with how we can improve Dec-LPMARL in the future. The following are the possible modifications that can be made in the future to enhance the performance of Dec-LPMARL in the SC2 environment.
> > >
> > >
> > > ***1. Stabilize the learning of decentralized policy***
> > >
> > > One of the main reasons for the performance degradation of Dec-LPMARL may be its training process because amortizing centralized policy using decentralized policy is not easy. To stabilize the training of Dec-LPMARL, we may need a regularizer that stabilizes the task allocation, such as a temporal regularization term that prevents the fluctuation of task allocation. Or we can add an additional communication layer to mediate the allocation result instead of learning a fully decentralized policy.
> > > We believe this approach may help Dec-LPMARL to get closer performance to LPMARL, which already outperforms other decentralized hierarchical MARL algorithms.
> > >
> > > ***2. Learning constraints of LP layer***
> > >
> > > As the constraints of the LP control the feasible solution space of the optimization problem, it plays a significant role in finding high-level action. However, we currently learn only the objective function coefficients of the task whose constraints are specified prior, and thus the constraint coefficients are fixed. Thanks to the differentiable optimization framework on QP [1], we can differentiate through the parameterized objective coefficient and the parametrized constraint coefficient. In this way, we can learn to dynamically impose intrinsic constraints for even tasks without explicitly imposed constraints. If the dynamic constraint coefficients are well-trained, the feasible solution space of high-level allocation may decrease; thus, Dec-LPMARL (and LPMARL) can be further stabilized.
> > >
> > > [1] Amos, Brandon, and J. Zico Kolter. "Optnet: Differentiable optimization as a layer in neural networks." International Conference on Machine Learning. PMLR, 2017.

---

> ### Author Response · Authors · 2022-11-15
> **Comment for the reviewer VRWi (2)**
>
> > The clarity of the paper generally needs improvement. For example, this paper uses (hierarchical) Dec-POMDP as the base framework, so each agent receives observation oi,t not the state st. However, LPMARL uses the state information (see Figure 1, where the input is the state) to construct the agent-task score, and it is unclear how the state can be constructed based on observations. Similarly, the paper uses si, but it should be oi. Lastly, Equation 8 uses the state to decide the low-level action, but it should be oi.
>
> **(State construction)**
> Thank you for pointing out the inconsistency of notations used in the manuscript. The global state $\boldsymbol{s} = \{s_k:k\in N\cup M \}$ is the collection of features for all the agents and tasks; for example, the locations, capacity, type, etc. Given the global state $\boldsymbol{s}$, we construct a directed graph $\mathcal{G}=(\mathcal{V,E})$ to describe the relationship between the agents and tasks explicitly. Then, the constructed graph $\mathcal{G}$ is used to construct the agent-task score.
>
> **(Input information scope for LPMARL and Dec-LPMARL)**
> LPMARL solves a task allocation problem to output the agent-task score in a centralized manner while using the global state s. Dec-LPMARL amortizes this central task allocation optimization using a decentralized task allocation model mapping the local observation o_i to task allocation probability; thus, Dec-LPMARL uses local observation $o_i=o_i (\boldsymbol{s}) $ for the global state $(\boldsymbol{s}) $. The information used by the two models during each decision-making step is summarized as follows:
>
> |            | High-level action                      | Low-level action                       |
> |------------|----------------------------------------|----------------------------------------|
> | LPMARL     | Global state $\boldsymbol{s}$          | Local observation $o_i$ for agent $i$  |
> | Dec-LPMARL | Local observation $o_i$ for agent $i$  | Local observation $o_i$ for agent $i$  |
>
> Because Figure 1 represents the decision-making procedure of LPMARL only, we have extended Figure 1 to include the decision-making procedure of Dec-LPMARL. In the revised figure, we clearly expressed the information used by LPMARL and Dec-LPMARL during each decision-making step.
>
> We also modified the following in the revised manuscript. The changes are marked with red font text.
> * In Dec-LPMARL, we use $o_i$ instead of $s_i$
> * We replace $\boldsymbol{s}$ to $o_i$ in Equation 8.

---

> ### Author Response · Authors · 2022-11-15
> **Comment for the reviewer VRWi (3)**
>
> > Regarding the non-hierarchical MARL baselines, it is unclear why the weighted sum of the high- and low-level rewards is used to train them. Wouldn't it be fairer to train them with the environment rewards only?
>
> We apologize for not clearly describing the reward conditions of the experiment in the paper. To clearly explain the reward setups, we included the following table in the Appendix C.1 of the revised manuscript:
>
> | Reward setting |            | Hierarchical MARL, LPMARL                                                   | Non-hierarchical MARL                       |
> |----------------|------------|-----------------------------------------------------------------------------|---------------------------------------------|
> | Dense          | high-level | +1 when all agent reaches landmark                                          | 0.5* high-level reward+0.5*low-level reward |
> |                | low-level  | +1 when each agent reaches landmark + distance and collision-related reward |                                             |
> | Sparse         | high-level | +1 when all agent reaches landmark                                          | 0.5* high-level reward+0.5*low-level reward |
> |                | low-level  | +1 when each agent reaches landmark                                         |                                             |
>
> LPMARL and Hierarchical MARL are designed to use sparse rewards that come only when an event (completing one or all the tasks) occurs. Because non-hierarchical MARL algorithms do not specify the reward setups, these can obviously utilize such sparse reward settings, albeit their performance is prone to degrade because these are not explicitly designed for sparse reward settings
>
> We wanted to investigate whether LPMARL and Hierarchical MARL algorithms can be trained well with dense reward settings as well. However, due to their inherent structure, LPMARL and hierarchical-MARL algorithms must use high-level reward (i.e., sparse reward) to train high-level policy. Thus, we use both sparse and dense rewards for LPMARL and Hierarchical MARL. Although Non-hierarchical MARL can be trained only with dense reward (distance and Collison-related reward), we use a weighted sum of the high-level (sparse) and low-level reward to make sure that two types of algorithms use the same reward information and thus compare the performance fairly.
>
> In case reviewers believe adding an additional sparse reward to non-hierarchical MARL is not beneficial, we have conducted additional experiments showing the effects of giving the additional sparse reward to non-hierarchical MARL. The results are shown in the following table. Giving the additional sparse reward rarely affects the performance; sometimes it improves the performance but sometimes not. We included the result of using only dense reward for non-hierarchical MARL in Appendix D.1.3.

---

### Official Review · Reviewer_vJe1 · 2022-10-25

**Confidence:** 3
**Correctness:** 4
**Technical Novelty And Significance:** 3
**Empirical Novelty And Significance:** 3
**Recommendation:** 6

**Clarity, Quality, Novelty And Reproducibility:**

The quality of this paper is fair. The clarity needs to be improved. The proposal of LP-based approach is novel to me.

**Strength And Weaknesses:**

Strength:
1. The proposed LP-based approach for agent-task allocation is novel to me. The paper discusses how the LP formulation can be generated using GNN and how to train it in an end-to-end manner with implicit function theorem.
2. The mathematical formulation in this paper is clearly written.

Weakness:
1. The motivation of this work is not clear to me. The problem definition, challenges, and background are not discussed in a clear manner.
2. It does not contain theoretical justification of why the proposed approach works as desired.

**Summary Of The Paper:**

This paper considers training a multi-agent reinforcement learning (MARL) model with sparse reward. It proposes a linear programming-based hierarchical MARL for the considered problem. It claims that the proposed approach learns an optimal agent-task allocation and also a local cooperative policy for agents in sub-groups.

**Summary Of The Review:**

The motivation of this paper is not clear to me. The problem definition, challenge, and background needs to be discussed in more detail. The proposed LP-based approach is novel and the paper provides a complete end-to-end training diagram for the proposed approach. However, it lacks theoretical justification of why it works as desired.

---

> ### Author Response · Authors · 2022-11-18
> **Comment for Reviewer vJe1 (1)**
>
> > Weakness 1: The motivation for this work is not clear to me. The problem definition, challenges, and background are not discussed in a clear manner.
>
> The target problem the current study aims to solve is the Decentralized Partial Observability Markov Decision Process (POMDP) with sparse reward. The problem definition is in Section 2.1. Solving Dec-POMDP is equivalent to deriving decentralized policies for agents such that individually chosen actions by decentralized policy induce coordination to maximize the success ratio (sparse reward). To clearly show this problem's definition and objective, we have modified Figure 1. It now shows the overall procedure of decision-making.
>
> The challenges in solving dec-POMDP with sparse reward are summarized as follows:
>
> •	Sparse reward: Because the reward is realized after a long sequence of actions, it is difficult to distribute the final reward over time and agents.
>
> •	Constraints: Even training a single-agent reinforcement learning policy for solving MDP with constraints is challenging.
>
> LPMARL performs better in sparse reward settings because LPMARL considers the agent-task relationships and explicitly solves agent-task allocation optimization problems in determining the lower-level action.

---

> ### Author Response · Authors · 2022-11-18
> **Comment for Reviewer vJe1 (2)**
>
> >Weakness 2: It does not contain a theoretical justification of why the proposed approach works as desired.
>
> We agree that the current study does not provide any theoretical analysis and justification for the proposed method. The target problem that the current problem seeks to solve belongs to Dec-POMDP or Cooperative Markov Game. Theoretical analysis is only allowed for a specific class of Dec-POMDP or Markov games. For example, the transition model is linear gaussian, and the reward function is a quadratic function of state and action.
>
> Instead of investigating the theoretical analysis of why the proposed approach works, we conducted ablation studies extensively investigating empirically why the proposed work.
>
>
> **<Effect of LP layer in decision-making procedure>**
>
> The high-level policy of LPMARL is composed mainly of two parts: (1) cost coefficient generating function producing the cost matrix to be used by LP agent-task allocation problem considering the relationships among agents and task, and (2) finding the solution of LP to allocate agents to tasks optimally.
>
> •	To evaluate the effectiveness of (1), we have compared LPMARL and LP (distance), the model using the cost coefficient matrix computed solely based on the distance between agents and tasks.
>
> •	To evaluate the effectiveness of (2), we have compared LPMARL and No-LP, the model that does not use LP in generating the agent-task allocation results but uses just the GNN layer to produce assignment probability.
>
> Figure 6 in Appendix D.1.2 shows the results of these ablation studies. LP (distance) using hand-designed features induces good coordination for tasks where proximity plays essential roles, such as cooperative navigation. However, devising a hand-defined rule for a complex task is challenging. In such cases, the proposed algorithm that constructs the cost matrix considering the global state can play an important role in deriving an effective policy. By comparing LPMARL and No-LP in tables 1, 2, 3, and 4, we can see that the LP layer plays a vital role in allocating agents to tasks to increase the success rate.
>
> **<Effect of end-to-end training>**
>
> we decompose the whole task into higher-level assignments and low-level local coordination among agents. We then have an LP layer to conduct the higher-level tasks and MARL to conduct the lower-level coordination tasks. We believe that this effective division of roles has contributed to performance improvement. Note that although we divided the decision-making roles, we train the high- and low-level policies in an end-to-end manner while backpropagating the MARL loss through the LP optimization layer using the implicit function theorem. As a result, these two modules work together to increase overall performance.

---

### Official Review · Reviewer_fFLH · 2022-10-25

**Confidence:** 3
**Correctness:** 3
**Technical Novelty And Significance:** 3
**Empirical Novelty And Significance:** 4
**Recommendation:** 6

**Clarity, Quality, Novelty And Reproducibility:**

quality : normal

clarity : good

originality : good. I liked the idea of taking this 3 step process "make a planning problem, solve the planning problem, execute the plan" into end-to-end and distributed.

evaluation : needs work. "interpretation" can be significantly expanded upon, "transferability" can be significantly expanded as well. at the bare minimum, show some starcraft playing videos.

**Strength And Weaknesses:**

strength : the technique is sound on a cursory glance (of the math), and the evaluation is thorough, showing (for the most part) good performance when compared with baseline algorithms.

weakness: I think the aspects of interpretability and transferability should be further explored. here's some comments:

1) on interpretability, the author only looked at a simple navigation environment where the interpretation is rather intuitive (or "obvious") -- distance is useful. I would like to see if a similar kind of intuition can be seen in the starcraft environment. The interpretability aspect is also not vetted against 3rd party users (i.e. crowd-workers) but rather the authors themselves, which lowers the validity of the "interpretability" claim. One could imagine a small survey on a crowd-worker setting where behaviours of several different algorithms were demonstrated, and the workers asked to describe them. Then, one can measure whether the descriptions agree with each-other as a proxy for interpretability. also, what about a video? I used to play a lot of starcraft and I'm actually in a good position to judge if the agents are behaving in a predictable way if only I can see a few videos, but all I get is success ratio, which hides away a lot of the details (did they succeed without taking any damage? did they do it fast or really dragged it out? did they focus fire? did they pull back the unit that was hurt? did they reposition themselves to a favorable flank before engaging? did protoss learn to abuse the fact that shields regenerate while HP do not?) these are the kind of "interpretation" that I would like to see.

Does the LP problem, invented by the neural network from end-to-end training, reflect these interpretable strategies? You can imagine an intervention, where the end-user is allowed to control these LP problems with some instrumentation, which should result in behavioural changes to the agents. This is similar in spirit to giving the user a slide-bar on some latent disentagled vector representation and have them slide it over. Can a similar experiment be constructed?

2) on transferability. what about 20m_vs_20m? what about 12 stalkers vs 24 zealots? there's actually quite a bit of combinatorial space of match ups, and presumeably fairly easy to set-up and run. You might discover the agents need to adopt different strategies after a certain "critical points" -- for instance, 2 stalkers can probably brute-force kill 2 zealots, but against 4 you really need to kite.

**Summary Of The Paper:**

multi-agent RL problem can be generally solved in 3 stages: 1) given the current state, a centralized planning problem is formulated, the problem is about which agent should do what (on a high level). 2) the planning problem is solved, allocating each agent with their respective high-level goal. 3) the agents individually carry out the high level goal using low-level controls.

this paper takes these 3 stages, and 1) make them end-to-end and 2) make it decentralized. the result is a model architecture and a training method with an inductive bias towards planning. the authors demonstrate that when agents are trained this way they often do better than the baselines.

**Summary Of The Review:**

overall, the paper is technically sound (as far as I could tell), and the quantitative results are comparable with baseline algorithms.

however, the qualitative claim on interpretation is bit of a stretch, as their notion of interpretability is essentially "the authors themselves looked at the resulting multi-agent strategy, and decided that 'yep, it made sense'". A 3rd party judgement would be good, extending the analysis on interpretability to starcraft would be good.

---

> ### Author Response · Authors · 2022-11-15
> **Response for the reviewer fFLH**
>
> >On interpretability, the author only looked at a simple navigation environment where the interpretation is rather intuitive (or "obvious") -- distance is useful. I would like to see if a similar kind of intuition can be seen in the starcraft environment. The interpretability aspect is also not vetted against 3rd party users (i.e. crowd-workers) but rather the authors themselves, which lowers the validity of the "interpretability" claim.
>
> Due to the limited time for preparing the rebuttal, we could not survey in a crowd-worker setting. Instead, we have made videos showing how the LP layer dynamically assigns each agent one of the tasks during StarCraft game playing. Please click the following link and watch the videos:
> [Link]( https://www.youtube.com/playlist?list=PLJC36nT1Y0NzuK8Nid9K_UhJYdz3Kwjuu).
>
> Videos show that the LPMARL assigns agents to enemies (tasks) considering not only the distance but also the health level.
>
> In addition, we have constructed Figure 7 composed of sequential snapshots of the video to show how the proposed model dynamically re-allocates the tasks to the agents. Each snapshot in Figure 7 corresponds to the moment where a new event occurs, i.e., an agent or an enemy dies. A detailed description is in Appendix D.2.
>
> ---
>
> >on transferability. what about 20m_vs_20m? what about 12 stalkers vs 24 zealots? there's actually quite a bit of combinatorial space of match ups, and presumeably fairly easy to set-up and run. You might discover the agents need to adopt different strategies after a certain "critical points" -- for instance, 2 stalkers can probably brute-force kill 2 zealots, but against 4 you really need to kite.
>
> We conducted transferability experiments only on the “conservative cooperative navigation task.” The primary objective of the transferability test is to investigate whether the trained cost-generating function can construct an LP for agents-tasks constrained allocation problems in unseen tasks with different agents and tasks. That is, for LPMARL, transferability implies the capability of formulating an LP over a new task. Please note that LPMARL achieves transferability for both the high-level LP constructing/solving and the low-level MARL policy.
>
> experiment results in Table 4 were obtained by the same environment setup; the policy is trained for a particular matchup, and the trained policy is tested with the same matchup (course, the initial locations of agents are all different). Most MARL algorithms were evaluated in a similar setup. There has rarely been a MARL algorithm that conducted a transferability test on the StarCraft environment because this environment is considered one of the most challenging benchmark cases for MARL.
>
> Respecting the reviewer’s suggestion, we have conducted additional experiments to test the transferability on StarCraft environment. Unfortunately, we have failed to achieve transferability in the StarCraft environment. We believe this is because there are more additional states to be considered during the game. In the future study, we will research ways to effectively achieve transferability in such complex hierarchical MARL environment.

---

> > ### Comment · Reviewer_fFLH · 2022-11-21
> > **doesn't seem to play starcraft well**
> >
> > I guess 8 marines should always beat 1+7 marines or 3+5 marines.
> > It doesn't look like the agent is pulling back hurt marines to de-aggro, instead letting it die.
> > It shows some level of focus firing, but cannot say for sure.
> >
> > All in all, I think if you just attack-moved the marines using the default behaviour you're likely to achieve similar results, since the red marines are clumped together to start, and should have advantage.
> >
> > I think a hand-crafted bot with some very simple potential field logic for conserving health, and a target acquisition preference based on %missing health on the enemy marine would reliably out-perform this trained agent, i.e. it should be able to beat MORE marines with less, i.e. it can probably win a 8 v 12 actually. (I'm saying this because I have implemented these hard-coded bots before)
> >
> > So I'm expecting performance of something like this (I just googled dragoon AI and got this) : https://youtu.be/x7LIFG-Hn8Q
> >
> > As a result I think this paper suffers from the issue of "the math is correct yet the artifact isn't really better than baseline". Where the baseline is just attack-move your marines.
> >
> > I appreciated the open-ness of the response and the inclusion of the video, and the comments on 20m vs 20m didn't work out. This is a hard problem. I'd like to thank the authors for making this engagement possible.

---

### Decision · Program_Chairs · 2023-01-20

**Decision:**

Reject

**Justification For Why Not Higher Score:**

Not sufficient support by reviewers. Comparison to baseline would have to be stronger and overall clarity of the paper needs to be improved.

**Justification For Why Not Lower Score:**

N/A

**Metareview: Summary, Strengths And Weaknesses:**

I thank the authors for their submission and active engagement during the discussion period.  This is a borderline paper.  On the positive side, reviewers acknowledge  that the method is sound [fFLH], novel [vJe1,VRWi], interesting [LMn4], clear [vJe1], and has the desired properties of interpretability [VRWi], and the empirical evaluation is thorough [fFLH,LMn4]. On the negative side, reviewers noted that comparison to baselines is not fair [VRWi], the clarity of the paper should be improved [vJe1,VRWi] and the method doesn't seem to beat baselines [fFLH]. Given this and the observation that the author's rebuttal doesn't seem to have addressed the concerns of reviewers fFLH and VRWi sufficiently, I believe the weaknesses currently outweigh the strengths of this paper. Therefore, I recommend rejection.